# TLCrys: Transfer Learning Based Method for Protein Crystallization Prediction

**DOI:** 10.3390/ijms23020972

**Published:** 2022-01-16

**Authors:** Chen Jin, Zhuangwei Shi, Chuanze Kang, Ken Lin, Han Zhang

**Affiliations:** 1College of Computer Science, Nankai University, Tianjin 300350, China; jinchen_cs@mail.nankai.edu.cn (C.J.); kangchuanze@mail.nankai.edu.cn (C.K.); 2College of Artificial Intelligence, Nankai University, Tianjin 300350, China; zwshi@mail.nankai.edu.cn (Z.S.); ken_lin@mail.nankai.edu.cn (K.L.)

**Keywords:** protein crystallization, transfer learning, attention mechanism, pre-training, fine-tuning

## Abstract

X-ray diffraction technique is one of the most common methods of ascertaining protein structures, yet only 2–10% of proteins can produce diffraction-quality crystals. Several computational methods have been proposed so far to predict protein crystallization. Nevertheless, the current state-of-the-art computational methods are limited by the scarcity of experimental data. Thus, the prediction accuracy of existing models hasn’t reached the ideal level. To address the problems above, we propose a novel transfer-learning-based framework for protein crystallization prediction, named TLCrys. The framework proceeds in two steps: pre-training and fine-tuning. The pre-training step adopts attention mechanism to extract both global and local information of the protein sequences. The representation learned from the pre-training step is regarded as knowledge to be transferred and fine-tuned to enhance the performance of crystalization prediction. During pre-training, TLCrys adopts a multi-task learning method, which not only improves the learning ability of protein encoding, but also enhances the robustness and generalization of protein representation. The multi-head self-attention layer guarantees that different levels of the protein representation can be extracted by the fine-tuned step. During transfer learning, the fine-tuning strategy used by TLCrys improves the task-specialized learning ability of the network. Our method outperforms all previous predictors significantly in five crystallization stages of prediction. Furthermore, the proposed methodology can be well generalized to other protein sequence classification tasks.

## 1. Introduction

The functions of a protein are largely determined by its three-dimensional structure. Therefore, analyzing the three-dimensional structure of proteins is of great significance for understanding the molecular mechanism of biological processes and studying the pathogenesis mechanism of diseases. Furthermore, it can also provide key information for the development and design of drugs for human diseases. At present, existing methods used to identify the three-dimensional structure of protein sequences are electron microscopy [1], Nuclear Magnetic Resonance (NMR) spectroscopy [2], and X-ray diffraction crystallography (X-ray diffraction measurement, XRD) [3]. Compared with NMR and electron microscopy, XRD has the advantages of easy implementation, short execution time, and low research cost. Therefore, XRD has become the most popular method at present, about 80% of protein structures in protein data bank (PDB) are obtained by XRD. In the experiments of protein crystallization, the main concern is on the importance of performing XRD experiment on a crystallizable protein. However, only 2–10% of proteins can produce diffraction-quality crystals [4,5,6]. Hence, experimenting with X-ray diffraction crystallography for proteins that cannot crystallize at the current experimental level, are costly and time-consuming. Therefore, it is important to develop accurate and efficient computational methods of forecasting whether a protein can crystallize at the current experimental level or not.

Early-stage computational methods are based on classical machine learning or statistical algorithms and mainly focused on feature extraction of protein sequences, such as OBScore [7], ParCrys [8], CrystalP2 [9], XtalPred [10], PPCPred [11], SCMCRYS [12], SVMCRYS [13], PredPPCrys I & II [14], Crysalis I & II [15]. These methods can be simply regarded as two-stage classification: (i) physio-chemical and structural feature selection and extraction, and (ii) classification with different machine learning algorithms using the extracted features. However, all of these methods require fussy physio-chemical and structural feature selection from the raw protein sequences, thus the performances of these methods are dependent on the quality of feature extraction. Hence, these models lack generalization and robustness.

Deep learning has been widely applied in bioinformatics [16,17,18]. Recently, some remarkable end-to-end deep learning frameworks [19,20] have been utilized for crystallization prediction. In comparison with traditional machine learning algorithms, the above methods integrate representation learning and model training in a unified architecture and does not need to extract features before modeling. In addition to the architectural advantages of the task design, existing deep learning models require a lot of labeled data to learn the information. However, compared with the protein database, the amount of crystallization label data for protein sequences is not large enough. Therefore, the performances of these models are still not satisfactory in real world applications.

Transfer learning [21] defines two domains: source domain *S* and target domain *T*. Learning task on source domain $T_{S}$ helps to improve the performance of learning task on target domain $T_{T}$ by transferring the $T_{S}$-learning knowledge to $T_{T}$. Knowledge learned from source domain can significantly enhance the robustness and generalization of target task. Recently, deep learning models with pre-training, such as Transformers [22] and BERT [23], have achieved remarkable success on natural language processing tasks. These models include two steps. Firstly, pre-training is adopted to initialize the network weights and learn the representations. Secondly, the downstream task is performed based on the pre-training step. Since protein sequences have many similarities with natural language, these transfer learning models are also suitable for modeling biological sequences. Unlabelled protein sequences implicitly contain significant structural and functional information. As a result, these pre-trained tasks learn representations [24,25] of protein, which can be used for transfer learning and help to achieve good performance in downstream tasks such as secondary structure prediction and interaction prediction [24,26].

In order to overcome the problems of insufficient labeling training data and inaccurate model prediction results, and to explore the internal correlation between protein sequence modeling and crystallization propensity, we propose a novel transfer learning based method for protein crystallization prediction, called TLCrys. The predictor consists of two models and processes, a protein representation pre-training step through a self-supervised multi-task model on protein sequence and Gene Ontology (GO) annotations, and a multi-head self-attention fine-tuning step on protein crystallization dataset with pre-trained parameters. In summary, our main contributions of this paper are as follows.

As far as we know, this is the first time that transfer learning is applied to the protein crystallization prediction task based on the protein sequences. Compared with traditional machine learning models, our model is more interpretable and our predictions are more accurate.Since protein sequences generally contain complex spatial structure, we use global attention module and multi-task learning to pre-train the self-supervised model of protein sequences.In fine-tuning step, we apply a multi-head self-attention layer to extract the different levels of protein features from the global representation space during pre-training step.

In order to verify the effectiveness of the protein representations of pre-training and fine-tuning steps, we design an end-to-end direct learning pipeline for comparison of TLCrys. We combine the pre training model and fine-tuning model, and use the crystallization data to train directly. Experiments indicate that TLCrys is superior to current state-of-the-art models. Besides, in order to prove the effectiveness and robustness of our model, we also carry out ablation experiments and case study.

## 2. Materials and Methods

### 2.1. Overview of Model

The training process of TLCrys model for protein crystallization prediction consists of two parts.

Learning task on source domain: self-supervised pre-training step for protein representation on protein sequences and Gene Ontology annotations.Learning task on target domain: supervised fine-tuning step on protein crystallization dataset with pre-trained parameters.

The representations learned in pre-training can be regarded as the knowledge of source domain. Transferring the representation as the input of fine-tuning step helps to improve the performance of the target task, i.e. protein crystalization prediction. The key component of these parts is the attention module.

### 2.2. The Attention Module

The attention module used in the pre-training process of TLCrys model was proposed by Brandes [27]. This attention module is similar to the self-attention module used in the Transformer model [22]. It is well known that protein structure is an extremely complex spatial structure that loses its spatial characteristics when it is expanded into a secondary structure. Two residues that are spatially close to each other are far apart in a two-dimensional sequence. So when we construct protein representations, we use attention modules to focus on global characteristics as well as local characteristics. As illustrated in Figure 1, the architecture of the attention module consists of dual parallel paths, one proceeds locally and the other proceeds globally. The local representations collected in the first path are 3D tensors of shape $B\times L\times d_{local}$ where *B* is the batch size, *L* is the sequence length in each mini-batch, and $d_{local}$ is the dimension of the local representations. The second path produces 2D tensors of shape $B\times d_{global}$ as global representations.

For the local representation path, the input sequences pass through two different types of 1D convolution layers, dilated convolution and classical convolution layer. Meanwhile, for the global representation path, the input annotations pass through the fully-connected layer, and then the broadcast layer. The broadcast layer is a fully-connected layer that transforms the $d_{global}$-channel global features into $d_{local}$-channel local features. Then, they are added up to the local representation path. Based on this structure, the global representations influence the local representations. These outputs are summed to the inputs of the regularization layer and then pass through the same structure again to generate the global representations. The local representations influence the global representations by global attention block.

We use global attention mechanism as special self-attention mechanism which focuses on position information of each other through each position vector, whereas the global attention quantifies position-wise attention weights to the local input features according to the global input vector. The global attention modules have two independent kinds of inputs, a global representation vector $x\in R^{d_{global}}$, and a local representation vector $R(S_{1},S_{2},\ldots ,S_{R}\in R^{d_{local}})$. It outputs a global representation vector $y(y\in R^{d_{value}})$, which can be calculated as:(1)$$y=\sum_{i=1}^{R}z_{i}v_{i},$$
where *z* is the attention values defined by:(2)$$z_{1},z_{2},\ldots z_{L}=Softmax\left\{\frac{\langle q,k_{i}\rangle }{\sqrt{d_{key}}}\right\}$$

The global query vectors *q*, the positional key vectors $k_{i}$ and local value *v* are three different vectors obtained by three linear transformations $W_{q},W_{k},W_{v}$ functions separately:(3)$$q=tanh\left(W_{q}x\right),\;\;k_{i}=tanh\left(W_{k}x\right),\;\;v_{i}=\sigma \left(W_{v}S_{i}\right),$$
where $W_{k}\in R^{d_{key}\times d_{local}}$, $W_{q}\in R^{d_{key}\times d_{global}}$, $W_{v}\in R^{d_{value}\times d_{local}}$ are trainable parameters and $\sqrt{d_{key}}$ is the scaling factor. σ denotes the sigmoid function.

To extract multi-view information in multiple representation sub-spaces simultaneously, a multi-head attention mechanism is applied, which aggregates features from multiple heads in parallel and aggregates these head outputs. Note that $d_{value}=d_{global}/n_{head}$, where $n_{head}$ is the total number of the attention heads. Compared with self attention mechanism, global attention mechanism has the following advantages.

We use dilated convolution to expand the receptive field and capture multi-scale context information.Global attention regards global inputs as the key vector. Therefore, the number of computation parameters performed by the model grows only linearly with the sequence length, while the standard self-attention calculates a position-to-position consistency matrix and the amount of parameters grows quadratically. This linear growth is also applicable to the computation complexity and the memory consumption of the model.Owing to the informational interaction between the local and global representations, we can obtain not only the local features of adjacent amino acids, but also the global features of protein sequencesThe model, based on the convolutional and global attention layers, is more efficient and stable than that relying on recurrent layers.

As illustrated in Figure 2a, we pre train the protein representation through the self-supervised dual-task method by protein sequence and Gene Ontology (GO) annotations, We use 26 characters and some special tags to represent the protein sequence, including 20 standard amino acids, Selenocysteine (U), an undefined amino-acid (X). Specially, we use OTHER to represent one of the other amino acids, and every sequence starts with START token and ends with END tags. Sequences shorter than the sequence length of the mini-batch are padded with PAD tags. GO annotations of each sequence are encoded as binary vectors.

The pre-training step of TLCrys is based on the global attention module. In the local path, the embedding layer can map the input protein sequences into the local representation vectors $R^{d_{local}}$. In the global path, we use a fully-connected layer to transfer the input binary annotations into global representation vectors $R^{d_{global}}$. The pre- training model includes six attention modules in series. The global representation and local representation of the previous module’s output are input into the next attention module to obtain a new global representation and local representation.

The pre-training dataset on protein sequences and GO annotations are extracted from UniRef90 (https://www.uniprot.org/uniref/) (accessed on 10 December 2021). To enhance the robustness of the model, we randomly replace 5% tokens with other tokens as noise. Furthermore, the inputted GO annotations are added with random noise by removing existing annotations by 25% probability or totally removed by 50%. In summary, the pre-training step is a dual-task where the model is expected to simultaneously recover both the protein sequence and its known GO annotations. The hyperparameter settings of the pre-training step are shown in Table 1.

### 2.3. Pre-Training

The loss function of the pre-training step consists of two parts: the categorical cross-entropy over the protein sequences and the binary cross-entropy over the GO annotations. It is defined as follows.
(4)$$L=-\sum_{i=1}^{l}S_{i}\cdot log\left({\hat{S}}_{i}\right)-\sum_{j=1}^{N}\left(A_{j}\cdot log\left({\hat{A}}_{j}\right)+\left(1-A_{j}\right)\cdot log\left(\left(1-{\hat{A}}_{j}\right)\right)\right),$$
where *l* is the sequence length, $S_{i}\in \left\{1,\ldots ,26\right\}$ is the sequence real tag at position *i*, ${\hat{S}}_{i}\in \left\{0,1\right\}$ is the predicted probability at position *i*. *N* is the the number of annotations, $A_{j}\in \left\{0,1\right\}$ is the sequence real annotations at position *j*, ${\hat{A}}_{j}\in 0,1$ is the predicted probability of annotations *j*.

### 2.4. Fine-Tuning

As depicted in Figure 2, we extract every pair of global representations from each attention module and concatenate them with the output global representation as whole global representation $L=concatenate[L_{1},L_{2}\ldots L_{n}]$. We designe a multi-head self-attention layer to extract the important information from each representation.

The self-attention mechanism transforms the concatenated feature vector *L* into three feature vectors, the Query (*Q*), Key (*K*), and Value (*V*) by three different linear mapping functions. As depicted in Figure 2, the weight assigned to each value is calculated as the dot-product of the query with the corresponding key:(5)$$Attention\left(Q,K,V\right)=Softmax\left(\frac{QK^{T}}{\sqrt{d_{k}}}\right)V,$$
where $\sqrt{d_{k}}$ is the scaling factor, $d_{k}$ is the dimension of the vector *K*, and T is the transpose operation. This operation is also called scaled dot-product attention [22]. The *Q*, *K* and *V* are obtained by three linear transformations with the same input separately: (6)$$Q=LW_{Q},\;\;K=LW_{K},\;\;V=LW_{V},$$
where $W_{Q},W_{K},W_{V}\in R^{d_{L}\times d_{k}}$ are trainable parameters and $d_{L}$ is the dimension of feature map.

To extract multi-view information in multiple representation sub-spaces simultaneously, a multi-head attention mechanism is applied, where each head is an independent scaled dot-product [22] attention module:(7)$$head_{i}=Attention\left(QW_{i}^{Q},KW_{i}^{K},VW_{i}^{V}\right),$$
(8)$$Multi\left(Q,W,V\right)=Concat\left(head_{i},\ldots ,head_{h}\right)W^{o},$$
where $QW_{i}^{Q}$, $KW_{i}^{K}$, $VW_{i}^{V}\in R^{D\times d_{ki}}$ are the linear transformation parameters same as those in Equation (Equation 3) and $W^{o}$ is the linear transformation parameters for aggregating the extracted information from different heads. Note that $d_{ki}=d_{k}/h$, where *h* is the total number of the attention heads, here h=6.

Our model adopts binary cross entropy loss function with a $l_{2}$ normalization. The regularized objective function L(θ) is calculated as follows.
(9)$$L\left(\theta \right)=-\sum_{i=1}^{N}\left[y^{i}log{\hat{y}}^{i}+\left(1-y^{i}\right)log\left(1-{\hat{y}}^{i}\right)\right]+\lambda {∥\theta ∥}_{2}^{2}.$$

Here ${\hat{y}}^{i}$ represents the predicted label of *n*th protein sequence, $y^{i}$ represents its corresponding crystallization prediction label, and *N* is the size of the training set, λ denotes a hyperparameter of $l_{2}$ regularization, and θ denotes all parameters of the model.

The model is trained using the Adam [28] optimizer, with mini-batch gradient descent to minimize the objective function. Initially, parameters of all layers in the pre-trained model are frozen, and only the newly added fully-connected layer is trained for up to 40 epochs. Then, all parameters are unfrozen and trained for up to 40 additional epochs. At the end, we train the model for 1 final epoch with a larger sequence length. Throughout all epochs, we reduce the learning rate on plateau and apply early stopping based on an independent validation set.

Early stopping is a useful learning strategy to overcome overfitting during training. Specifically, when the validation loss is no longer decreasing for limited epochs, the training procedure terminates. The models of all epochs are then evaluated on the validation set and the one with the best performance is chosen as the final prediction model.

### 2.5. Direct Learning

In order to verify the effectiveness of the protein representation in pre-training and fine-tuning steps, we build an end-to-end pipeline that combines the two steps. To compare with the transfer learning process of TLCrys, the hyperparameter settings of the direct learning model are consistent with those of the previous model. The training strategy of direct learning also remains unchanged compared with the fine-tuning process. We construct an all-zero vector as the GO annotation input since our crystallization dataset lacks the GO annotations.

## 3. Results

### 3.1. Dataset and Environment

As we can see in Table 2, the experimental dataset of this paper is from the PredPPCrys [14] model (https://doi.org/10.1371/journal.pone.0105902.s007) (accessed on 10 December 2021), which includes five datasets in the form of FASTA. Each entry of the FASTA data consists of a protein residue sequence and a tag. The five corresponding tags include Sequence Cloning failed, Production of protein material failed, Purification failed, Crystallization failed, and Crystallizable. They respectively indicate final states of protein crystallization experiments.

The types of tags indicate the propensity of protein in different stages of crystallization: the protein cloning failure (CLF), the production of protein material failure (MF), protein purification failure (PF), and crystallization failure (CF) and crystallographic (CRYs) of final diffraction mass. In CLF task, only the cloning failed tag is negative. In MF task, Sequence Cloning failed, production of protein material failed are negative. In protein material production tendency task (MF), Sequence Cloning failed, production of protein material failed are negative. In the task of purification tendency (PF), purification failed and crystallization failed are negative, crystallizable is positive, and crystallizable is negative in the task of final diffraction mass crystallization. The numbers of training sets and test sets are shown in Table 2, in which the sequence similarity of test sets is 40%.

Our codes are implemented in Tensorflow which is a powerful deep learning framework. Trainable weight matrices of TLCrys are initialized by the default setting. TLCrys is trained on a single NVIDIA GeForce GTX 3070 GPU with 8GB memory.

### 3.2. Metrics

We evaluate our model on the dataset described in Section 3.1. Accuracy (ACC), Matthew’s Correlation Coefficient (MCC), balanced F1 Score, sensitivity (SENS), specificity (SPEC) and precision(PRE) are commonly used metrics for binary classification. All of them are based on the number of true positive (TP), true negative (TN), false positive (FP), and false negative (FN). They are defined as follows.
(10)$$Accuracy\left(ACC\right)=\frac{TP+TN}{TP+TN+FP+FN}$$
(11)$$Specificity\left(SPEC\right)=\frac{TN}{TN+FP}$$
(12)$$Sensitivity\left(SEN\right)=\frac{TP}{TP+FN}$$
(13)$$Precision\left(PRE\right)=\frac{TP}{TP+FP}$$
(14)$$F1-Score=\frac{2*Precision*Sensitivity}{Precision+Sensitivity}$$
(15)$$MCC=\frac{TP*TN-FP*FN}{\sqrt{\left(TP+FP)(TP+FN)(TN+FP)(TN+FN\right)}}$$

Accuracy is the most common evaluation metric, which means the proportion of correctly classified samples in all samples. Specificity is the ratio of correctly predicted negative samples. Sensitivity denotes the ratio of the positive samples that were correctly predicted. Precision represents the proportion of samples that are classified as positive samples that are actually positive samples. F1-score is a comprehensive evaluation metric and the harmonic average of precision and sensitivity. Matthew’s Correlation Coefficient is a remarkable metric in binary classification problem on imbalanced data [29].

Furthermore, the receiver operating characteristic (ROC) curve denotes the classification performance of a model by plotting the true-positive (TP) rate against the false-positive (FP) rate. TP rate and FP rate change when the different discrimination thresholds are selected. The area under ROC curve (AUR) is an important indicator for measuring the classification performance of a model.

### 3.3. Comparison with Other Methods

We evaluate predictive performances of 13 models based on five stages of crystallization prediction dataset. They are OBScore [7], XtalPred [10], CrystalP2 [9], Crysalis I & II [15], PredPPCrys I & II [14], ParCrys [8], TargetCrys [30], PPCPred [11], SVMCRYS [13], SCMCRYS [12] and DeepCrystal [19]. Here, only Crysalis I & II model and PredPPCrys I & II model can predict the classification results of the five stages. The five-stage classification of protein crystallization is specifically the sequence failing to clone (CLF), protein material production failed (MF), protein purification failure (PF), and crystallization failure (CF) and diffraction-quality crystals (CRYs). According to the statistical data in Table 3, the performance of TLCrys in the five-stage protein crystallization prediction is better than all the previous predictors. Some metrics, such as SPEC or SEN, are related to the classification tendency and do not represent the comprehensive performance of the classifier. In terms of comprehensive metrics such as AUC, MCC, and F1-Score, TLCrys is significantly better than other compared models.

### 3.4. Ablation Experiments

In order to verify the effectiveness of the multi-head self-attention layer in fine-tuning module and select the number of heads required, we conduct ablation experiments. Firstly, we remove the multi-head self-attention layer and directly send the concatenated whole representations into the fully-connected layer for the output layer. Then, we set up different numbers of heads to determine the best number of heads. As we can see from Table 4, when we set a 6-head self-attention layer, the model has the best performance in comprehensive metrics. From the experimental results, we can conclude that the multi-head self-attention layer of TLCrys can extract important information from each representation, which can improve the accuracy and robustness of the model.

### 3.5. Case Study

Transcription factors are defined as some sequence-specific proteins, which can regulate many essential biological processes. Sox transcription factors consist of highly conserved high-mobility group (HMG) domain of 70∼80 amino acids [31]. In Sox transcription factor family, Sox9 is a gene that can target several important organs, such as the brain, heart, kidney, and bone. Sox17 can participate in endoderm differentiation in early mammalian development [20]. Five protein sequences of Sox9 and Sox17 are applied to validate the performance of TLCrys and other predictors. The recent research has illustrated that Sox9 HMG, Sox17 HMG and Sox17EK HMG are competent to conduct diffraction-quality crystallization, while it is not evident that full-length sequences of Sox9 and Sox17 (i.e., Sox9 FL and Sox17 FL) are competent to produce diffraction-quality crystals [19,31,32]. Therefore, an excellent protein crystallization predictor should output low probability scores while processing Sox9 FL and Sox17 FL, and output high probability scores while processing Sox9 HMG, Sox17 HMG and Sox17EK HMG.

Table 5 shows the predicted probability score of the TLCrys and other predictors for Sox transcription factor proteins. Here we use 0.5 as a threshold, if the score is more than 0.5, the protein is predicted to be crystalizable. TLCrys and DeepCrystal correctly identifies all the proteins that are able to produce diffraction-quality crystals. However, compared with DeepCrystal, TLCrys achieves lower probability prediction scores while processing full length sequences of Sox9 and Sox17, and achieves higher probability prediction scores while processing Sox9 HMG, Sox17 HMG and Sox17EK HMG. The results suggest that TLCrys is a more credible protein crystallization predictor compared with current predictors.

## 4. Conclusions

Crystallization prediction of protein is a significant task in computational biology. Due to the profound relationship between protein crystallization and protein structure, we design a novel transfer learning based method for protein crystallization prediction, named TLCrys. In source domain, TLCrys adopts a multi-task training method in pre-training procedure to obtain global and local information of protein sequences for protein representation learning. In target domain, the representations are regarded as knowledge from the source domain to enhance the fine-tuning model for protein crystalization prediction. Besides, the multi-head self-attention mechanism is adopted in fine-tuning. Through the comparative experiment of transfer learning and direct learning, the profound relationship between protein crystallization and protein structure is revealed. The ablation experiments demonstrate the effectiveness and number of attention modules in the fine-tuning model. Case study validate the capability of our method for protein crystallization prediction. The experiments demonstrate that our method significantly outperforms other methods on five crystallization stages of prediction on test sets. The proposed methodology is generally applicable and can be used to address any other sequence classification tasks.

## Figures and Tables

**Figure 1 ijms-23-00972-f001:**
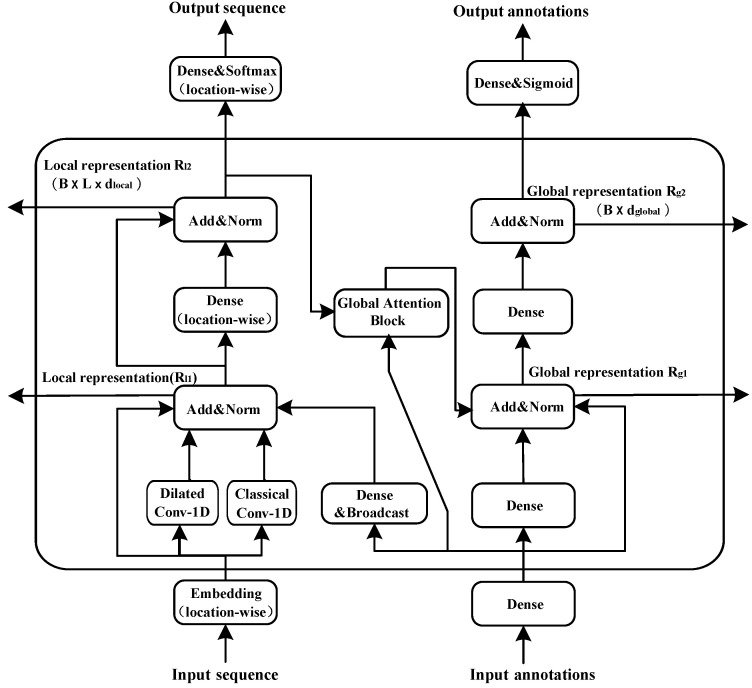
Structure of attention module in TLCrys pre-training step.

**Figure 2 ijms-23-00972-f002:**
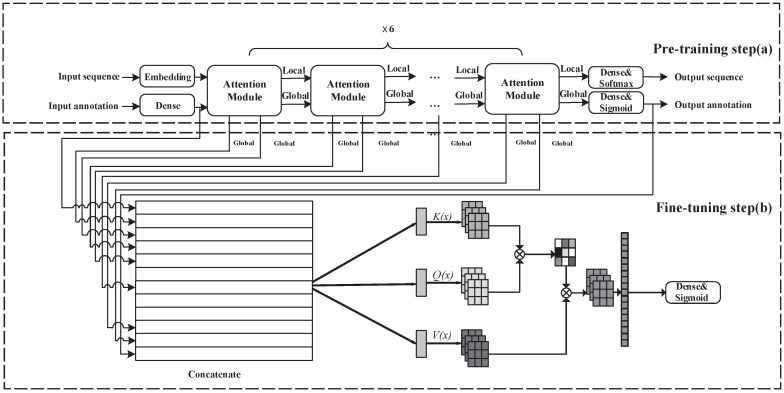
The architecture of TLCrys consists of two parts: (**a**) self-supervised pre-training protein representation models on protein sequences and Gene Ontology annotations. (**b**) supervised fine-tuning on protein crystallization dataset with pre-trained parameters.

**Table 1 ijms-23-00972-t001:** Parameters of TLCrys.

**Global Dim**	**Local Dim**	**Dilation Rate**	**Kernel Size**
512	128	5	9
**Stride Size**	**Key Dim**	**Value Dim**	**Head Number**
1	64	128	4

**Table 2 ijms-23-00972-t002:** Statistics of datasets

Tasks	Dataset	Clone f.	Material Production f.	Purification f.	Crystallization f.	Crystallization
CLF	Train	N:9502	P:14428			
	Test	N:1939	P:2852			
MF	Train	N:17017		P:6913		
	Test	N:3347		P:1444		
PF	Train	-		N:2318	P:4702	
	Test	-		N:474	P:932	
CF	Train	-			N:224	P:631
	Test	-			N:35	P:138
CRYs	Train	N:19509				P:4421
	Test	N:3892				P:899

**Table 3 ijms-23-00972-t003:** Comparison of our two model with other methods on test sets.

	Model	AUC	MCC	ACC (%)	SPEC (%)	SEN (%)	PRE (%)	F1 Score (%)
CLF	PredPPCrys I	0.711	0.296	65.33	63.58	66.50	73.16	69.67
	PredPPCrys II	0.725	0.322	66.54	65.56	67.20	74.44	70.63
	Crysalis I	0.731	0.332	66.98	66.60	67.22	75.56	71.15
	Crysalis II	0.756	0.365	68.34	69.95	68.34	76.85	72.35
	Direct learning	0.701	0.326	64.65	76.14	56.84	77.81	65.69
	TLCrys	0.817	0.455	72.90	74.28	71.96	80.46	77.00
MF	PredPPCrys I	0.772	0.380	69.93	68.21	72.88	49.95	59.27
	PredPPCrys II	0.793	0.416	71.95	71.36	73.30	52.70	61.32
	Crysalis I	0.759	0.377	70.23	69.93	70.99	49.25	58.15
	Crysalis II	0.793	0.427	73.08	73.58	73.09	54.15	62.21
	Direct learning	0.745	0.307	73.31	88.67	37.74	58.98	46.03
	TLCrys	0.848	0.446	78.37	92.53	45.57	72.47	55.90
PF	PredPPCrys I	0.800	0.460	74.83	70.52	77.02	83.77	80.25
	PredPPCrys II	0.872	0.579	79.73	81.43	78.86	89.31	83.76
	Crysalis I	0.796	0.436	73.87	67.80	73.87	82.47	77.93
	Crysalis II	0.793	0.427	73.08	73.58	73.09	54.15	62.21
	Direct learning	0.778	0.505	78.52	60.97	73.09	54.15	62.21
	TLCrys	0.861	0.583	81.58	70.25	87.34	85.24	86.27
CF	PredPPCrys I	0.712	0.280	67.05	67.65	66.91	89.42	76.54
	PredPPCrys II	0.735	0.175	69.47	68.89	69.50	97.80	81.26
	Crysalis I	0.739	0.281	65.50	70.59	64.23	89.80	74.89
	Crysalis II	0.752	0.337	62.57	85.29	56.93	93.97	70.90
	Direct learning	0.694	0.123	71.10	31.43	81.16	82.35	81.75
	TLCrys	0.785	0.459	79.77	68.57	82.61	91.20	86.69
CRYs	ParCrys	0.611	0.132	59.66	60.56	55.91	25.40	34.93
	OBScore	0.638	0.184	59.28	57.78	65.49	27.14	38.38
	CRYSTAP2	0.599	0.123	51.64	48.10	67.78	22.28	33.54
	XtalPred	-	0.224	65.04	65.61	62.51	29.31	39.91
	SVMCRYs	-	0.142	55.11	52.78	65.70	23.39	34.50
	PPCPred	0.704	0.254	63.63	62.09	70.67	29.03	41.15
	XtalPred-RF	-	0.205	60.94	59.67	66.41	27.56	38.95
	SCMCRYS	-	0.145	60.93	62.01	56.24	25.48	35.07
	PredPPCrys I	0.770	0.326	69.65	69.30	71.13	35.23	47.12
	PredPPCrys II	0.838	0.428	76.04	76.21	75.30	42.64	54.45
	Crysalis I	0.788	0.339	71.00	70.89	71.41	35.50	47.42
	Crysalis II	0.838	0.435	76.27	76.28	76.20	42.84	54.85
	DeepCrystal	0.858	0.477	77.83	77.43	79.51	45.90	58.20
	Direct learning	0.801	0.367	83.79	95.99	31.03	64.14	41.83
	TLCrys	0.879	0.546	87.24	94.96	53.84	71.18	61.30

Area Under Curve (AUC), Matthew’s Correlation Coefficient (MCC), Accuracy (ACC), Specificity (SPEC), Precision(PRE), Sensitivity (SENS), and Precision (PRE).

**Table 4 ijms-23-00972-t004:** Ablation experiments on CRYs task.

Models	AUC	MCC	ACC (%)	F1 Score (%)
without multi-head attention	0.874	0.498	86.57	54.36
2 heads	0.875	0.537	86.69	61.32
4 heads	0.880	0.529	86.99	59.09
TLCrys (6 heads)	0.879	0.546	87.24	61.30
8 heads	0.879	0.541	87.30	60.10

**Table 5 ijms-23-00972-t005:** Predicted probability value of the TLCrys and other predictors for Sox transcription factor proteins.

Model	Sox9 FL (−)	Sox9 HMG (+)	Sox17 FL (−)	Sox17 HMG (+)	Sox17 EK-HMG (+)
TLCrys	0.156	0.674	0.260	0.791	0.681
DeepCrystal	0.315	0.676	0.430	0.643	0.633
TargetCrys	0.032	0.045	0.037	0.029	0.031
Crysalis II	0.474	0.55	0.474	0.553	0.555
Crysalis I	0.438	0.482	0.487	0.567	0.557
PPCPred	0.039	0.658	0.089	0.462	0.523
CrystalP2	0.327	0.459	0.470	0.436	0.402

“+” represents crystallizable protein and “−” represents non-crystallizable protein.

## Data Availability

The pre-training dataset on protein sequences and GO annotations are extracted from UniRef90 (https://www.uniprot.org/uniref/) (accessed on 10 December 2021). The experimental dataset of this paper is from PredPPCrys (https://doi.org/10.1371/journal.pone.0105902.s007) (accessed on 10 December 2021).

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
