# Peer review of "TLCrys: Transfer Learning Based Method for Protein Crystallization Prediction"

_ijms, 2022, doi:10.3390/ijms23020972_

Round 1

Reviewer 1 Report

The manuscript proposes an ambitious task. A few minor comments may be made:

  1. Table 3: this might be easier to follow if the abbrevaiations were defined in the Table caption/legend.
  2. The manuscript is somewhat excessively cryptic when it comes to case studies where the performance of the newly proposed algorythms/procedures would be illustrated in straightforward manner. Up until Table 5 there aer several parameters defined for testing purposes - e.g. the ones in the first column in Table 3, not to mention the ones in the first line of the same Table. By contrast, Table 5, with the case study, does not clearly state which parameters were calculated and how many of them are compared across
  3. An extensive Supporting Information addition might be instructive/helpful for potential readers/users, mostly in view of comment 2 (above)

Reviewer 2 Report

The paper can be interesting for researchers working with protein structures.

Some aspects of the presentation can be improved. Indications are given here below.

KEYWORDS

*** It would be advisable to have at least five keywords. Keywords are important for potential readers to find the article on navigating.

*** “attention-based” is not a self-standing keyword, because it does not contain any noun

Line 32-37

“However, in the experiments of protein crystallization, XRD often fails in the multi-step process of crystallization…..”

*** This concept relates to features that appear to be crucial to understand the following parts. It would be advisable to explain what would be expected by XRD for the various steps of the crystallization process. Is it actually expected/desired to obtain information related to different steps. Or is the main focus on the importance of performing XRD experiments on a crystallised structure?

*** The meaning of “non-crystallization protein structure” is also not completely clear, because “crystallization” is a process, not the outcome of the process. Is the intended meaning “non-crystallised protein structure”?

Line 38

“forecasting protein crystallization”

*** It would be advisable to explain what exactly one wants to forecast: specific changes during the process plus the final result, or only the final result? In the former case, it would be important to briefly recall the aspects/changes that one wants to forecast.

LANGUAGE

Some corrections are needed, but they can be easily identified during copy-editing. Few examples are shown here below for illustration. I do not give examples concerning the use of articles, but I would like to note that it needs verification in a number of cases.

Line 48: “these models are lack of generalization and robustness” should be written as “these models lack generalization and robustness”.

Line 59. In the part “Learning task on source domain TS helps to improve the performance of that on target”, it is not clear to what “of that” is grammatically referred.

Lines 111-112: In the part “The attention module constituting to the TLCrys model is proposed by Brandes [27], which is similar to the transformer model [21].”

** “constituting to” is not correct. The intended meaning is not clear.

** in “which is similar”, it is not clear to what “which” is grammatically referred, and the intended meaning is not clear.

Line 271: In the part “correctly classified samples in all samples. the specificity represents the proportion of”, it is not clear whether the first sentence should end after “in all samples” (in which case the following “the” should be capitalised) or is meant to continue (in which case a comma should replace the full-stop).
